# Unveiling Neuroprotection and Regeneration Mechanisms in Optic Nerve Injury: Insight from Neural Progenitor Cell Therapy with Focus on Vps35 and Syntaxin12

**DOI:** 10.3390/cells12192412

**Published:** 2023-10-06

**Authors:** Hyun-Ah Shin, Mira Park, Hey Jin Lee, Van-An Duong, Hyun-Mun Kim, Dong-Youn Hwang, Hookeun Lee, Helen Lew

**Affiliations:** 1Department of Biomedical Science, CHA University, Pocheon-si 13488, Gyeonggi-Do, Republic of Korea; sha9547@naver.com (H.-A.S.); kyn0708@naver.com (H.-M.K.); hdy@cha.ac.kr (D.-Y.H.); 2Department of Ophthalmology, CHA Medical Center, CHA University, Pocheon-si 13488, Gyeonggi-Do, Republic of Korea; hoohoo99@chamc.co.kr; 3CHA Advanced Research Institute, CHA University, Pocheon-si 13488, Gyeonggi-Do, Republic of Korea; 4378nm@chamc.co.kr; 4Gachon Institute of Pharmaceutical Sciences, Gachon College of Pharmacy, Gachon University, Incheon 21936, Republic of Korea; duongand08@gmail.com (V.-A.D.); hklee@gachon.ac.kr (H.L.); 5Department of Microbiology, School of Medicine, CHA University, Pocheon-si 13488, Gyeonggi-Do, Republic of Korea

**Keywords:** mitochondria, neural progenitor cells, optic nerve compression, optic nerve crush, Vps35, Syntaxin12

## Abstract

Axonal degeneration resulting from optic nerve damage can lead to the progressive death of retinal ganglion cells (RGCs), culminating in irreversible vision loss. We contrasted two methods for inducing optic nerve damage: optic nerve compression (ONCo) and optic nerve crush (ONCr). These were assessed for their respective merits in simulating traumatic optic neuropathies and neurodegeneration. We also administered neural progenitor cells (NPCs) into the subtenon space to validate their potential in mitigating optic nerve damage. Our findings indicate that both ONCo and ONCr successfully induced optic nerve damage, as shown by increases in ischemia and expression of genes linked to neuronal regeneration. Post NPC injection, recovery in the expression of neuronal regeneration-related genes was more pronounced in the ONCo model than in the ONCr model, while inflammation-related gene expression saw a better recovery in ONCr. In addition, the proteomic analysis of R28 cells in hypoxic conditions identified Vps35 and Syntaxin12 genes. Vps35 preserved the mitochondrial function in ONCo, while Syntaxin12 appeared to restrain inflammation via the Wnt/β-catenin signaling pathway in ONCr. NPCs managed to restore damaged RGCs by elevating neuroprotection factors and controlling inflammation through mitochondrial homeostasis and Wnt/β-catenin signaling in hypoxia-injured R28 cells and in both animal models. Our results suggest that ischemic injury and crush injury cause optic nerve damage via different mechanisms, which can be effectively simulated using ONCo and ONCr, respectively. Moreover, cell-based therapies such as NPCs may offer promising avenues for treating various optic neuropathies, including ischemic and crush injuries.

## 1. Introduction

Damage to the optic nerve can lead to axonal degeneration, which in turn results in a gradual decline in retinal ganglion cells (RGCs) and ultimately irreversible vision loss [1]. Such optic neuropathy can stem from various factors, including blunt head trauma, ischemia, metabolic disorders, genetic mitochondrial diseases, autoimmune inflammation, and infiltrative conditions [2]. The optic nerve crush (ONCr) model, involving a deliberate crushing of the optic nerve to induce RGCs apoptosis, has been broadly utilized to study neuronal death and survival mechanisms, as well as to evaluate potential therapeutic strategies for various types of optic neuropathy [3,4]. This model has been used in conjunction with pharmacological and molecular approaches to test and identify therapeutic agents. Previous studies have proven the efficacy of cellular treatments in the ONCr model, created using a self-clamping protocol with forceps [5,6]. Notably, this model has facilitated research into RGCs regeneration [7]. Thus, our previous studies also demonstrated the effectiveness of cellular treatment using the ONCr model [8,9].

In recent studies, the optic nerve compression (ONCo) injury model has emerged as a less hazardous alternative to the ONCr model, due to its minimal disruption of the ophthalmic artery and its blood flow. This method involves the use of Carlson DSEK graft forceps as opposed to self-clamping to temper the severity of the crush [10,11]. Investigations into the optic nerve transection (ONT) and ONCo have shed light on the origin and migration of myeloid and microglial cells in the optic nerve and optic nerve head (ONH). ONT, which uses scissors to sever the nerve while maintaining blood flow to the retina, is more effective than ONCo for inducing RGCs axon loss and degeneration in the retina. The latter leaves approximately 50% of the axons intact, allowing the corresponding RGCs to survive [11]. Moreover, the higher concentration of microglia and myeloid cell proliferation in the optic nerve compared to ONT suggests that a complete transection of the optic nerve may impede the migration of reactive myeloid cells to the retina [11].

To the best of our knowledge, no study to date has performed a comparative analysis of different types of optic nerve injuries. Therefore, we examined the patterns of damage and recovery using two distinct models of optic nerve injury, both of which avoid full transection of the optic nerve. We compared the ONCo and ONCr models to evaluate changes in essential mitochondrial function, RGCs regeneration, and axoplasmic flow in the optic nerve before and after injury, considering the injury pressure severity. In addition, we investigated the therapeutic potential of neural progenitor cells (NPCs) in both types of optic nerve injury.

## 2. Materials and Methods

### 2.1. Cell Culture and Hypoxic Damage Conditioning

Immortalized R28 retinal precursor cells were cultured as previously described [9]. H9 human ESCs (WiCell Research Institute, Madison, WI, USA) were routinely maintained on Matrigel-coated culture dishes (BD Biosciences, San Jose, CA, USA) in TeSR™-E8™ medium (STEMCELL Technologies, Vancouver, BC, Canada). The culture process for NPCs is detailed in a recent publication [8]. A hypoxic environment was created in R28 cells using cobalt chloride (CoCl_2_). R28 cells were seeded at a density of 2 × 10^5^ and then treated with 300 µM CoCl_2_ for 3 h. Following this, NPCs were added to the damaged R28 cells. After 24 h, the cells were harvested and prepared for analysis.

### 2.2. Construction of the Optic Nerve Compression (ONCo) and Crush (ONCr) Models

Six-week-old female Sprague Dawley (SD) rats (Koatech, Gyeonggido, Republic of Korea) were housed in standard animal facilities where food and water were provided at constant temperatures of 21 °C. All animal experiments were conducted in accordance with protocols approved by the Institutional Animal Care and Use Committee of Bundang CHA Medical Center (IACUC No. 220101). ONCo and ONCr were performed in the left eye (oculus sinister; OS) and the right eye (oculus dexter; OD) was used as control. The rats were anesthetized via an intraperitoneal injection of Zoletil and Rompun. A lateral canthotomy and conjunctival incision were made after the topical application of 0.5% proparacaine hydrochloride. Subsequently, the tissues surrounding the optic nerve were carefully dissected to expose the optic nerve without damaging the adjacent blood supply. The ONCo model was constructed using Calison DSEK graft forceps, 3 ¼”, non-self-closing (Ambler Surgical, Exton, PA, USA) to apply mild compression to the nerve 2 mm behind the globe for 5 s [10,11]. Conversely, the ONCr model was created by applying strong crush pressure using extra-fine self-closing forceps. After thoroughly suturing the canthal site, a subtenon injection of NPCs was performed on the nasal side of the eyeballs. The animals were euthanized after 1, 2, and 4 weeks, and the tissue was collected for analysis. The rats were grouped as follows: the Sham group (received a balanced salt solution [BSS] injection after optic nerve compression) and the NPC group (received a 2 × 10^6^/0.06 mL injection after optic nerve compression).

### 2.3. Immunoblot Analysis

Total proteins from cells or at least three of tissues were extracted using either RIPA or PRO-PREP buffer (iNtRON Biotechnology, Gyeonggi-do, Republic of Korea). Protein concentration was determined using the BCA method (Thermo Fisher Scientific, Waltham, MA, USA). Target proteins were separated via SDS-PAGE and then transferred to PVDF membranes (GE Healthcare, Chicago, IL, USA). The membranes were incubated overnight at 4 °C with primary antibodies (Appendix A). After a series of washing steps, the membranes were incubated with horseradish peroxidase-conjugated anti-rabbit or mouse secondary antibodies at 1:10,000 dilution (GeneTex), again overnight at 4 °C. Target bands were visualized using enhanced chemiluminescence solutions (Bio-Rad Laboratories, Hercules, CA, USA) and were detected on an ImageQuant™ LAS 4000 (GE Healthcare, Chicago, IL, USA).

### 2.4. Proteomics

The comprehensive process for proteomics analysis is described in a previous paper [9]. Briefly, proteomic analyses were conducted on R28 cells treated with PBS (control), with CoCl_2_, and NPCs to elucidate the extensive effects of NPCs on undamaged cells. Protein digestion was performed using the filter-aided sample preparation (FASP) protocol [12] with Ultracel^®^ YM-30 centrifugal filters (Merck Millipore, Darmstadt, Germany).

Sample analysis was performed on an LC-MS/MS system, specifically a Dionex Ultimate 3000 HPLC coupled with a Q Exactive™ Hybrid Quadrupole-Orbitrap MS (Thermo Fisher Scientific). This was equipped with an Acclaim™ PepMap™ 100 C18 nano-trap column (75 μm × 2 cm, 3 μm particles, 100 Å pores, Thermo Fisher Scientific), and an Acclaim™ PepMap™ C18 100A RSLC nano-column (75 μm × 50 cm, 2 μm particles, 100 Å pores, Thermo Fisher Scientific, Waltham, MA, USA). The sample loading flow rate was 2.5 μL/min. Peptide mixtures were separated with solvent A and solvent B (0.1% FA/80% acetonitrile) at a flow rate of 300 nL/min. The gradient setup for solvent B was as follows: 4% (14 min), 4–20% (61 min), 20–50% (81 min), 50–96% (1 min), 96% (10 min), 96–4% B (1 min), and 4% (17 min). The nano-electrospray ionization source was operated in positive mode with a spray voltage of 2.0 kV. Additional parameters included a capillary temperature of 320 °C, an isolation width of ± 2 *m*/*z*, a scan range of 400–2000 *m*/*z*, and resolutions in full-MS scans and MS/MS scans (at 200 *m*/*z*) of 70,000 and 17,500, respectively. MS was performed using a data-dependent acquisition method. The top ten precursor ions with the highest intensity were isolated in the quadrupole and fragmented by higher-energy collisional dissociation with 27% normalized collisional energy. Dynamic exclusion was set at 20 s to minimize repeated analyses of the same abundant precursor ions.

### 2.5. Data Processing and Bioinformatics

Raw MS/MS data files were analyzed against a SwissProt human protein database using Proteome Discoverer (Version 2.4) equipped with Sequest HT. The search parameters were set as follows: 10 ppm for precursor ion mass tolerances, 0.02 Da for fragment ion mass, and a maximum of 2 missed cleavages with the trypsin enzyme. Peptide sequence modifications included static carbamidomethylation of cysteine (+57.012 Da), dynamic modifications of methionine oxidation (+15.995 Da), carbamylation of protein at the N-terminal (+43.006 Da), and acetylation of protein at the N-terminal (+42.011 Da). A false discovery rate (FDR) cutoff of 1% was applied. Statistical comparisons of protein abundances across groups were made using the Student’s t-test. Differentially expressed proteins (DEPs) were filtered with a cutoff *p*-value ≤ 0.05 and log2FC ≥ 1 (fold-change). A heatmap was generated using Proteome Discoverer. Protein–protein interactions were analyzed using the STRING database (https://string-db.org/ (accessed on 06/06/2023). Volcano plots were prepared using R software (Version 3.6.1). Gene ontology analysis was performed using Database for Annotation, Visualization and Integrated Discovery (DAVID) [13].

### 2.6. Immunofluorescence of Retinal Whole Mounts and RGC Survival Analysis

To prepare retinal whole mounts, both normal and injured eyes were immersion-fixed overnight in 4% paraformaldehyde (PFA), after which the retina was isolated. The retina was sectioned into four parts to allow it to lay flat on the slide. Then, the whole mounts were rinsed three times for 15 min each in phosphate-buffered saline (PBS; pH 7.4). For enhanced permeability, the retinas were frozen for 15 min at −80 °C in PBS/0.5% Triton^®^ X-100. The retinas were incubated with primary antibodies overnight at 4 °C (Appendix A), and after rinsing, they were incubated with Alexa Fluor conjugated secondary antibodies (Rabbit, Alexa 555, Invitrogen, Waltham, MA, USA) for 2 h at room temperature. DAPI was used as a counterstain.

Then, the retinas were transferred onto a slide, with the inner retina facing upwards, using a paintbrush. PBS was removed and a mounting medium (Dako, Glostrup, Denmark) was added. A small piece of broken coverslip glass was placed between the slide and coverslip glass to prevent crushing and damaging the retina. After the coverslip was positioned, images were captured using confocal microscopy (LSM 880; Carl Zeiss, Jena, Germany) for fluorescence quantification.

### 2.7. Optic Nerve Histological Analysis

Optic nerves were fixed with 4% PFA, dehydrated with ethanol, and embedded in paraffin. For histological analysis, paraffin blocks were sectioned into 5 μm slices and Hematoxylin and Eosin (H&E) staining was performed using standard methods. To conduct immunofluorescence staining, sections were deparaffinized in xylene and rehydrated in ethanol. For antigen retrieval, slides were heated in a microwave using 0.01 M citric acid buffer (pH 6.0) (Biosesang, Gyeonggi-do, Republic of Korea) for 10 min, and subsequently rinsed with PBS. Then, the slides were incubated with a blocking solution (Protein Block Serum-Free, Dako) for 1 h. Subsequently, slides were incubated with primary antibodies overnight at 4 °C. The following day, the samples were incubated with secondary antibodies conjugated with Alexa Fluor 488 and 555 (Invitrogen) for 1 h at room temperature. DAPI was used as a counterstain, followed by mounting with a fluorescence mounting medium (Dako). Images were captured using an Axio slide scanner and processed using ZEN 3.1 Software.

### 2.8. Small-Interfering RNA for Vps35 Protein

For the knockdown of the VPS35 protein, we used a target sequence of siRNA supplied by Precaregene (Seoul, Republic of Korea). The sequence used for siRNA rat VPS35 was 5′- ACA GUG GAG AUA UUC AAU AAA CUT A. As a negative control, we used siRNA Negative Control (scramble) provided by Precaregene. R28 cells were transfected using Lipofectamine 3000 (Thermo Fisher Scientific, Waltham, MA, USA) in accordance with the manufacturer’s instructions.

### 2.9. Statistical Analyses

All results are presented as the mean ± standard error of the mean (SEM). Data analyses were performed using GraphPad Prism 9 software (GraphPad Software, Inc., La Jolla, CA, USA). The statistical significance criteria used for data analyses are detailed in the figure legends.

## 3. Results

### 3.1. Comparison of Optic Nerve Injury Models

To compare the two models, we injured the optic nerve 2 mm behind the globe using two types of forceps (Figure 1A). Then, we analyzed the regulation of Hif-1α, Vegf, Prdx2, Nlrp3, Tnfα, and IL-6 protein expression in the retinas of rats at 1, 2, and 4 weeks post-injury (Figure 1B). At 1 and 2 weeks, hypoxia-involved proteins (Hif-1α and Vegf) and inflammation-related proteins (Nlrp3, Tnfα, and IL6) showed significant increases in the injured retina. In particular, Hif-1α and Nlrp3 proteins were induced in the ONCo injury group. IL-6 induction was significant in the ONCr injury group, which could have caused the Prdx2 reduction at 2 weeks post-ONCr. To assess the effects of NPCs, we examined the expressions of hypoxia-, microglial-, neuronal-, and inflammation-related proteins in NPC-injected retinas (Figure 1C). In the ONCo injury group after NPC injection, inductions of Hif-1α significantly decreased, and Iba-1 and Thy-1 recovered along with Bdnf induction at 4 weeks post-ONCo. The reduction of Tnfα and IL6 was dominantly observed in the ONCr injury group. This decrease in inflammation-related proteins could be linked to the recovery of neuronal markers such as NeuN, Nf, and Gap43. 

### 3.2. NPCs Protect Retinal RGCs in Optic Nerve Injury Models

We examined the survival rates of RGCs in retinas following hypoxia injury. We calculated rates of recovery by counting cells that tested positive for target proteins in whole mounts stained with Brn-3a and Tuj1 at 1, 2, and 4 weeks post-optic nerve injury and NPC injection (Figure 2A). At 1 week, we observed a significant increase in Tuj1 expression (2.35- and 1.6-fold, respectively) following NPC injection in both types of injury (Figure 2B). However, Brn-3a expression did not recover within 1 week post-NPC injection. In the 2-week group, compared to the 1-week period, there was a significant 1.47-fold increase in Brn3a-positive cells with ONCo alone. The count of Tuj1-positive cells tended to decrease at 2 weeks post-injection, but this was not statistically significant. Furthermore, at 4 weeks post-NPC injection, only Brn-3a expression dramatically recovered, showing a 3.26-fold increase in the ONCo injury group.

### 3.3. Effects of NPCs on Optic Nerve Axon Recovery in Each Model

We investigated the damaged areas in each group to evaluate the expression of neuron markers in injured optic nerves. Compared to the 1-week sham group of ONCo, the injured area was larger by 4.67-fold in the ONCr group (Figure 2C). After NPC injections, all injured areas trended towards recovery over time. In particular, at 4 weeks post-ONCo, the injured area significantly decreased by 60% following NPC injection. The 2-week ONCr group also displayed a 49% reduction in the injured area. In these groups, we observed that recovery mediated by NPC was faster than spontaneous recovery.

To evaluate the protective effects of NPC injection on optic nerves, we calculated Gap43- and Gfap-positive cells in the optic nerves of hypoxia-damaged models (Figure 2D,E). We discovered a significant increase (1.74-fold) in the expression of Gap43 at 4 weeks in the ONCo model (Figure 2D). Conversely, in the ONCr model, NPCs demonstrated a recovery effect (1.49- and 1.71-fold, respectively) early post-injection (1 and 2 weeks). We also confirmed that Gfap expression in the optic nerve did not recover in either group (Figure 2E).

### 3.4. Hierarchical Clustering and Gene Ontology

We performed a proteomic analysis of R28 cells. Hierarchical clustering of differentially expressed proteins (DEPs) was conducted under three conditions: Control, CoCl_2_, and CoCl_2_+NPC (Figure 3A). Compared to the control group, the CoCl_2_ group exhibited downregulation of 97 proteins and upregulation of 120 proteins (Figure 3B). In addition, GO analysis of the DEPs was also performed using DAVID. Using a false discovery rate ≤0.1, the GO functional clusters were enriched and categorized (Table 1).

### 3.5. Identification of Regulated Target Proteins during Recovery by NPC in Hypoxia-Damaged Retinal Precursor Cells

We utilized volcano plot analysis to identify changes in protein expression when hypoxic cells were exposed to NPC (Figure 3C). We discovered that Syntaxin12, Delphilin, and Vps35L expression were significantly increased in CoCl_2_-exposed cells following NPC exposure (compared to CoCl_2_ alone). In the STRING database, we observed interactions between Syntaxin12, Vps35L, and Vps35 proteins (Figure 3D).

### 3.6. Role of VPS35 and Recovery of Mitochondrial Function Induced by NPCs

To investigate whether NPCs promote cellular recovery under hypoxic conditions, we exposed R28 cells to CoCl_2_ for 3 h, and then co-cultured them with NPCs. When the R28 cells were exposed to CoCl_2_, the expression of Hif-1α significantly increased by 2.5-fold. However, co-culturing with NPCs reduced Hif-1α expression by 0.44-fold. In contrast to Hif-1α, the expression of mitochondrial homeostasis-related proteins such as Vps35, Mul1, Mfn2, Mff, Lonp1, P62, Lc3b, Prdx2, and Prdx5 were reduced by factors of 0.12-, 0.24-, 0.17-, 0.14-, 0.26-, 0.32-, 0.35-, 0.45-, and 0.3-fold, respectively, upon exposure to CoCl_2_. Most of these reductions were recovered by NPCs (Figure 4A). We also verified the expression of Vps35, Lonp1, P62, Lc3b, Prdx2, and Prdx5 in the retinas of optic nerve-damaged animal models (Figure 4B). Interestingly, the expression levels of these proteins were significantly increased by NPC injection in the ONCo group, with the exception of Prdx2. This was further confirmed in a replicated in vitro Vps35 knockdown study (Figure 4C). In Vps35 deficient cells, mitochondrial homeostasis-related proteins Mul1, Mfn2, Mff, Lonp1, P62, Lc3b, Prdx2, and Atp5a were significantly reduced by 0.57-, 0.83-, 0.82-, 0.80-, 0.80-, 0.83-, 0.84-, and 0.87-fold, respectively, compared with the scramble_control. All of these proteins were decreased by CoCl_2_-induced damage; NPCs, however, could not recover the reduction without Vps35.

Based on these results, we suggest that NPCs have a role in maintaining mitochondrial quality control through the Vps35 protein in ONCo animal models.

### 3.7. Identification of Syntaxin12, a New Target Protein Regulated by NPCs during RGCs Recovery

We were able to identify novel proteins, Syntaxin12 and Delphilin, in hypoxia-damaged R28 cells after co-culturing with NPCs, as indicated by our proteomics data (Figure 5A). In addition to in vitro analysis, Syntaxin12 and Delphilin expression in the retina was significantly increased only in the ONCr 4-week group, not in the ONCo group, following NPC injection (Figure 5B). We also investigated the signaling pathways involved in retinal tissues during regulation by NPCs. We found that the Vps45 protein and the Wnt signaling pathway were involved in the RGCs recovery process facilitated by NPCs. Vps45, Wnt3a, p-Gsk3β, and Lef1 were increased in ONCr groups, with the exception of p-β-catenin. We evaluated the density of Syntaxin12 expression in the injured areas of the optic nerves for both groups (Figure 5C). Following NPCs injections, Syntaxin12 significantly increased at 1 week in the ONCr group (by 1.28-fold) and at 2 weeks (by 1.7-fold) in the ONCo group. Similar to the sham groups, both the ONCo and ONCr groups displayed reductions in Syntaxin12 density in the injured area (66%, 41%, 36%, and 30%, respectively) at the 2- and 4-week time points.

## 4. Discussion

The optic nerve injury model necessitates meticulous construction to prevent excessive force or extended crushing, as these can damage the ophthalmic artery and induce retinal ischemia. Furthermore, extreme care is needed to avoid harming other blood vessels surrounding the animal’s eye. The level of precision required renders this model challenging to implement in research.

The ONCo model, which entails a compression injury, can progressively reduce blood flow to the optic nerve, leading to ischemia and nerve tissue damage. This form of injury can trigger axonal degeneration, myelin loss, and eventually the death of RGCs. Although a compression injury may still permit some nutrients and organelles to traverse the compressed region, it can disrupt axonal transport due to nerve deformation [8]. In comparison, a crush injury completely severs the axons, causing the total cessation of axonal transport and subsequent degeneration of the axons and neighboring cells. Therefore, it is anticipated that axoplasmic flow in the optic nerve would be more severely disrupted in crush injuries than in compression injuries.

The ONCr model, conversely, involves physical harm to the optic nerve due to excessive pressure or force. This type of injury can cause immediate axonal damage and the swift loss of RGCs [14]. Furthermore, the mechanical force of the crush injury can lead to tissue distortion and swelling, resulting in additional inflammation and fibrosis [15]. 

VPS35 is a component of the retromer complex, which plays an instrumental role in the endosomal sorting and trafficking of cargo proteins [16]. While its precise role in a hypoxic injury to neuronal tissue remains to be fully elucidated, certain evidence points towards a possible role in mediating cellular responses to hypoxia. For instance, studies have shown that reducing VPS35 expression in neuronal cells can impede the autophagic flux and increase vulnerability to hypoxic injury [17]. In addition, a VPS35 deficiency can intensify hypoxic-ischemic brain injury in animal models [18].

VPS35 also plays a role in the transport and preservation of mitochondria in neurons [19]. Specifically, it is associated with the retrograde transport of damaged mitochondria from axons to the cell body for degradation and recycling. This process is vital for maintaining mitochondrial health and avoiding the accumulation of damaged mitochondria [20]. Mutations or dysfunction in VPS35 have been connected to impaired mitochondrial transport and clearance, leading to mitochondrial dysfunction and neuronal degeneration [21]. However, further research is necessary to thoroughly understand the function of VPS35 in the context of a hypoxic injury to the neuronal tissue.

In this study, we also identified a novel target, Syntaxin12, which is regulated by NPCs during the RGC recovery process. Syntaxin12 belongs to the SNARE (soluble NSF attachment protein receptor) protein family, which plays a crucial role in vesicle fusion events during intracellular trafficking [22]. This protein localizes to endosomal compartments in various cell types and is involved in the process of vesicle fusion between synaptic vesicles and the presynaptic membrane during neurotransmitter release [23].

Dysfunction of Syntaxin12 has been linked with neurodegenerative diseases, including Alzheimer’s and Parkinson’s diseases [24]. Syntaxin16, another member of the same family, has also been implicated in the regulation of autophagy, a process by which damaged cellular components are degraded and recycled [25]. Therefore, understanding the role of Syntaxin12 in response to NPCs during the RGC recovery process could provide valuable insights for potential therapeutic strategies.

Evidence suggests that Syntaxin12 may play a role in the Wnt signaling pathway. One study demonstrated that Syntaxin12 interacts with Dishevelled-3 (Dvl3), a pivotal component of the Wnt signaling pathway [26]. This interaction is critical for the proper localization of Dvl3 to the plasma membrane and for the activation of downstream Wnt signaling events, which may have implications for the regulation of Wnt signaling in the neuronal tissue. However, further research is required to fully comprehend the relationship between Syntaxin12 and the Wnt signaling pathway in neuronal tissues.

Concerning the regulation of mitochondrial dynamics in neurons, some evidence suggests that Syntaxin4 interacts with proteins implicated in mitochondrial fission and fusion, such as dynamin-related protein 1 (Drp1) and Syntaxin17 with Mfn1, respectively [27,28]. In addition, the knockdown of Syntaxin1A impairs the mitochondrial function and leads to increased oxidative stress in neurons [29]. To gain deeper insights into the role of Syntaxin or Delphilin in neural processes and their associated signaling mechanisms, the utilization of a cell-based biohybrid sensor device for chemical source direction estimation becomes instrumental. This innovative device not only aids in the exploration of cell functionality but also provides directional guidance towards chemical sources within retina progenitor cells like R28 cells, facilitating the precise material analysis involved in signal transmission [30]. Recently, it was found that Syntaxin12 and the COMMD3/CCC complex interact physically and functionally with disease-associated proteins VPS33B and VPS16B in the α-granule biogenesis of platelets [22]. These findings suggest that Syntaxin12 may play a role in maintaining the health and proper functioning of mitochondria in neurons.

Various studies have indicated a relationship between VPS35 and the Wnt pathway. The Wnt pathway plays a crucial role in several biological processes, including neuronal development and synaptic plasticity. VPS35 participates in the Wnt signaling pathway by regulating the degradation of Wnt receptors and modulating the activity of β-catenin, a key downstream effector of the Wnt pathway [31,32]. In particular, it is involved in the endocytosis and recycling of Wnt receptors, including Frizzled and LRP6, which are necessary for Wnt signaling [33]. These studies suggest that VPS35 plays a role in regulating the Wnt pathway in neuronal tissue. However, further research is needed to fully understand the relationship between hypoxia and VPS35 in nerve tissues.

There is limited research on the specific impact of inflammation on VPS35 in nerves, but inflammation could potentially influence the VPS35 function in other cell types. Moreover, studies have demonstrated that inflammation-induced oxidative stress can lead to VPS35 dysfunction, contributing to the pathogenesis of neurodegenerative diseases such as Parkinson’s disease [34]. Thus, it is plausible that inflammation might also affect the function of VPS35 in nerves. However, more research is required to fully elucidate this relationship.

Recent research has shed light on the critical role of neuronal immuno-metabolism in the recovery process following an optic nerve injury. The interaction between neuronal metabolism and immune responses is bidirectional. And neuronal-derived metabolites could modulate the activation state of microglia and macrophages. This crosstalk is critical in determining whether the immune response following optic nerve injury is conducive to tissue repair or exacerbates damage [35,36,37]. For example, excessive availability of glucose in diabetes activates the pro-inflammatory M1 macrophages, innate lymphoid cells, and T and B cells. When these cells bind to their respective receptors on neurons, it can lead to an upregulation of pro-inflammatory cytokines, resulting in sustained neuronal excitotoxicity and subsequent neuronal loss [38]. Furthermore, semaphorins, vital for nervous system development, play a pivotal role in regulating immune responses and metabolic disorders as axon guidance molecules [39]. Notably, the Sema3A-Nrp1 signaling pathway has been identified for its significant involvement in the pathogenesis of diabetic retinopathy [40,41]. Additionally, Sema4D has demonstrated a protective effect against impaired wound healing in diabetic conditions [42]. Moreover, disruptions in the Sema3C-Nrp2 axis due to diabetes have been linked to corneal dysfunction, leading to issues such as delayed wound healing and impaired nerve regeneration [43]. Therefore, Sema3A-Nrp1 signaling could affect neuronal energy metabolism and mitochondrial function, and dysregulation of these processes can influence the survival and regenerative capacity of injured neurons.

Inflammation can be triggered by various stimuli, which in turn activates intracellular signaling pathways such as NF-κB and MAPK. Other pathways and mechanisms might also be involved under certain conditions, such as autoimmune diseases, viral infections, and cancer. For instance, the release of TNFα might be triggered by Toll-like receptors (TLRs) or other pattern recognition receptors (PRRs) on immune cells, as well as cytokine signaling and other pathways [44]. In addition, the activation of intracellular protein complexes known as inflammasomes can also instigate inflammation. Inflammasomes are typically activated by pathogen-associated molecular patterns (PAMPs) or damage-associated molecular patterns (DAMPs), which stimulate cytosolic pattern recognition receptors (PRRs), such as NOD-like receptors (NLRs) [45]. This can result in the activation of caspase-1, leading to a robust inflammatory response by cleaving and activating pro-inflammatory cytokines such as IL-1β and IL-18 [46]. However, hypoxia counteracts inflammation through the downregulation of the binding of mTOR and NLRP3 and the activation of autophagy, which is protective in mouse models of colitis [47]. This study significantly contributes to the field of molecular neurodegeneration research, offering novel insights and potential therapeutic strategies for neural injuries and related degenerative conditions. And it paves the way for future investigations and translational applications, providing a foundation for developing targeted therapies that harness the regenerative power of neural progenitor cells and the identified proteins.

## 5. Conclusions

In this study, we compared the ONCo and ONCr models to investigate changes in essential mitochondrial function and axoplasmic flow in RGCs in the optic nerve during and after damage caused by different degrees of injury pressure. We also explored the potential therapeutic mechanisms of NPCs in both types of optic nerve injury. Our findings suggest that NPCs could serve as an effective cellular treatment for various optic neuropathies, and that the recovery pathways vary depending on the extent of the optic nerve damage.

## Figures and Tables

**Figure 1 cells-12-02412-f001:**
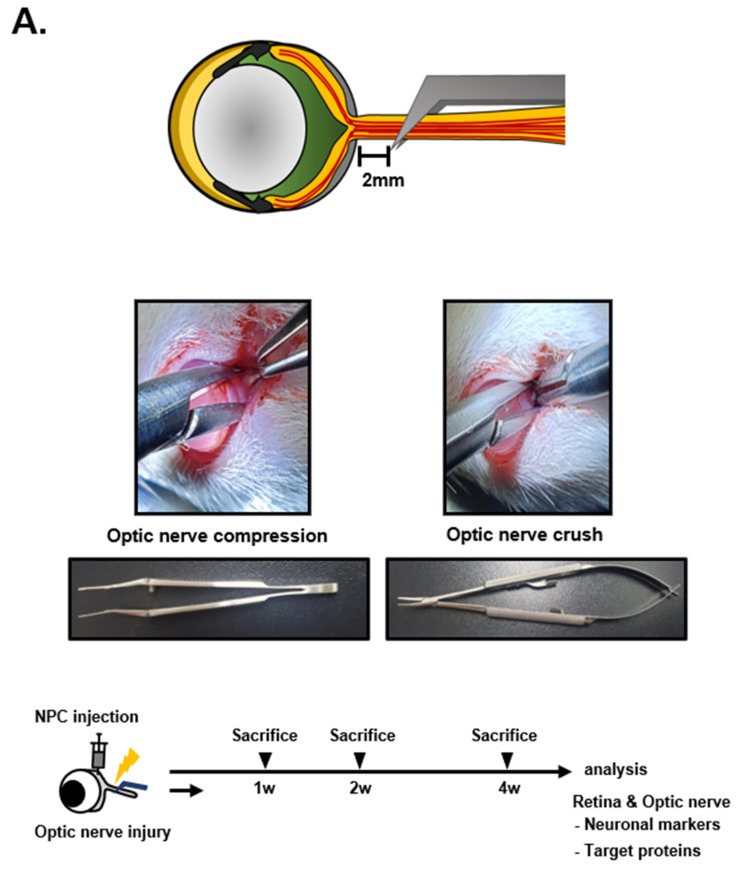
Analysis of ONCo and ONCr in vivo models. (**A**) The scheme of in vivo optic nerve injury model construction. Changes in target proteins were assessed by immunoblot analyses of rat retina extracts. The samples were analyzed 1, 2, and 4 weeks after injection with optic nerve injury. (**B**) Comparison with two types of optic nerve disease models. Quantified values of Hif-1α, Vegf, Prdx2, Nlrp3, Tnfα, and IL6 expression in OS compared with OD are presented. (**C**) After neural progenitor cell (NPC) injection, the retinas were also analyzed. Expression levels were normalized to β-actin and the values of OS were divided OD. The results are presented as the mean ± standard error of the mean (SEM) of the independent retina and optic nerve analyses and are expressed as fold changes compared to the control. Significant differences were estimated using an unpaired *t* test (* *p* < 0.05, ** *p* < 0.01, *** *p* < 0.001 vs. the age-matched sham [BSS]). OD, oculus dexter; OS, oculus sinister.

**Figure 2 cells-12-02412-f002:**
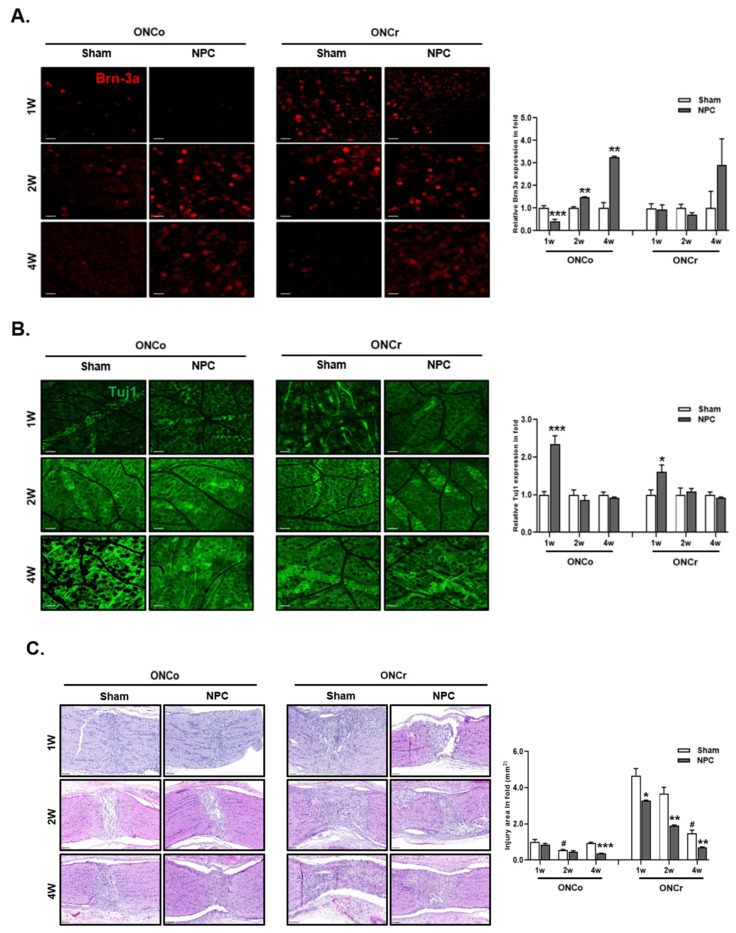
NPC promote retinal ganglion cell and axon regeneration in the optic nerve injury animal model. Representative confocal microscopy-based fluorescence images following (**A**) Brn-3a and (**B**) Tuj1 staining (original magnification: ×400) of NPC injection in the optic nerve-injured animal model (scale bar: 20 μm). (**C**) H&E stain of an injured area on the optic nerve (scale bar: 100 μm). (**D**) Gap43 and (**E**) Gfap fluorescence quantification were measured in a box with 7690 mm^2^ including the injured area (scale bar: 100 μm). Total Gap43 and Gfap positive cells were measured using ZEN software. Total of two retinas and optic nerves from each group were used. The results are presented as the mean ± SEM. Significant difference was estimated using an unpaired *t* test (* *p* < 0.05, ** *p* < 0.01, *** *p* < 0.001 vs. the age-matched sham; # *p* < 0.05 vs. 1 w sham of each group).

**Figure 3 cells-12-02412-f003:**
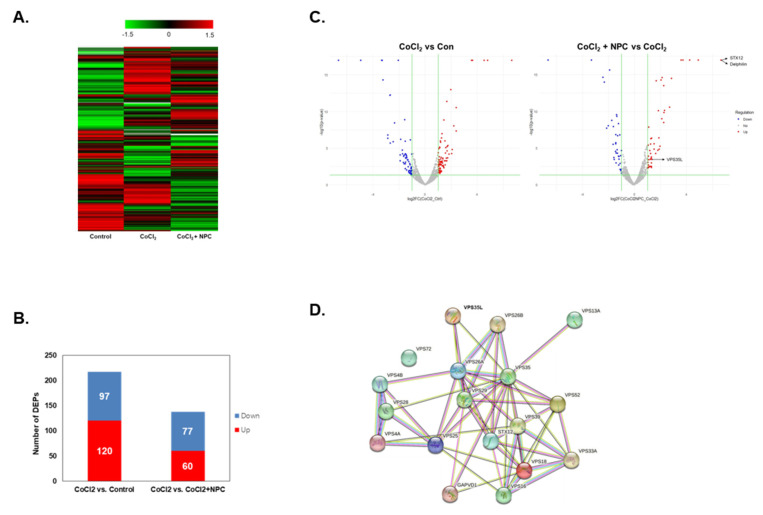
Cluster analysis of DFEs by NPCs in hypoxia-damaged R28 cells. Identification of differentially expressed proteins in treated groups (CoCl_2_, and CoCl_2_ + NPC) and PBStreated control cells. (**A**) Heatmap of differentially expressed proteins (*p* ≤ 0.05 and log_2_FC ≥ 1) in lysates of three groups (Control, CoCl_2_, and CoCl_2_ + NPC) analyzed by hierarchical clustering. High expression is shown in red; low expression is shown in green. (**B**) The graph showing the number of up and down-expressed proteins identified in proteomic analysis of each group in R28 cells. (**C**) Volcano plot of DEPs of each group. Red dots describe up-regulated proteins and blue dots represent down-regulated proteins. Proteins with significantly different expressions are presented. (**D**) Identification of protein-protein interaction of DEF in experimental groups of R28 cells.

**Figure 4 cells-12-02412-f004:**
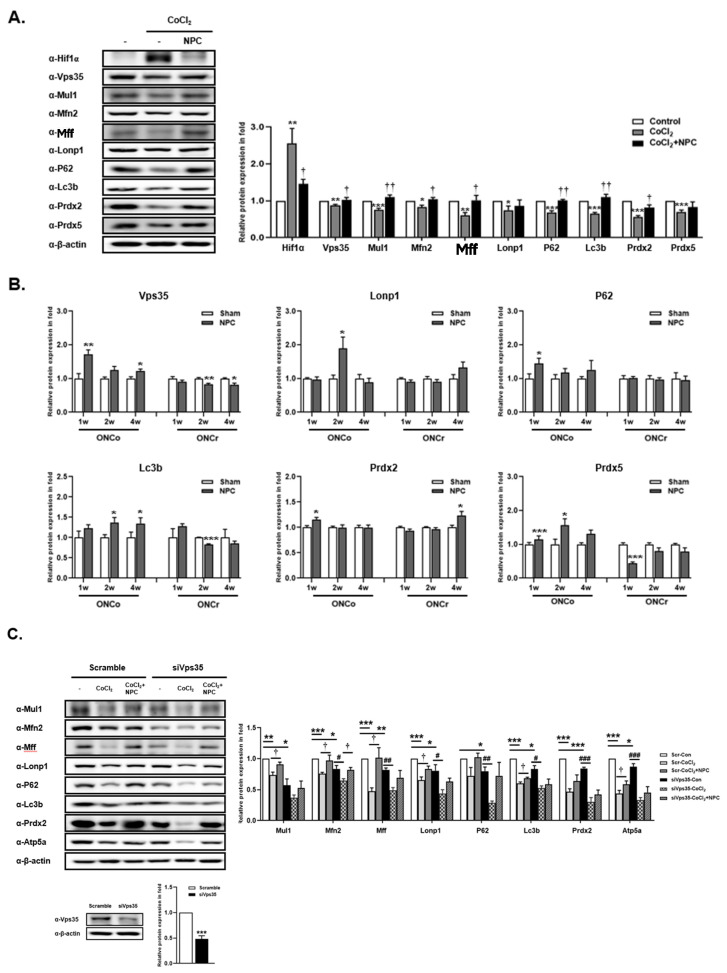
NPCs have a function to rescue mitochondria quality control of hypoxic damaged in vitro/in vivo models. The R28 cells were treated with CoCl_2_ (300 μM). Then, R28 cells were cultured with NPCs after 3 h of CoCl_2_ treatment. (**A**) Mitochondrial homeostasis-related protein expressions were also determined after 24 h. (**B**) Using in vivo retina tissues, immunoblot analyses of mitochondrial-regulated protein expression levels were also evaluated. (**C**) Scramble and siRNA targeting Vps35 were transfected into R28 cells. After incubation, Vps35 deficient cells were exposed to CoCl_2_ (300 μM). After incubation for 3 h, the hypoxia-induced cells were co-cultured with NPCs. Then, Western blot analyses were performed. The results are presented as the mean ± SEM. Significant difference was estimated using an unpaired *t* test (* *p* < 0.05, ** *p* < 0.01, *** *p* < 0.001 vs. the control or the age-matched sham or scramble_control; # *p* < 0.05, ## *p* < 0.01, ### *p* < 0.001 vs. siVps35_control; † *p* < 0.05; †† *p* < 0.01vs. the CoCl_2_ of each group).

**Figure 5 cells-12-02412-f005:**
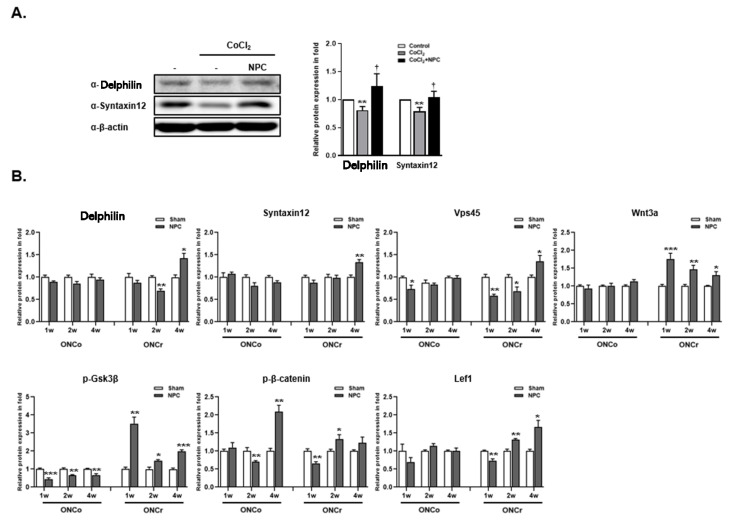
NPCs recover neuro-regeneration of hypoxia damaged in vitro/in vivo models. The R28 cells were treated with CoCl_2_ (300 μM). Then, R28 cells were cultured with NPCs after 3 h of CoCl_2_ treatment. (**A**) Delphilin and Syntaxin12 protein expressions were also determined after 24 h. (** *p* < 0.01 vs. control; † *p* < 0.05 vs. CoCl_2_). (**B**) Using in vivo retina tissues, immunoblot analyses of target protein expression levels were also evaluated. (**C**) Syntaxin12 fluorescence quantification was measured in the injured area (scale bar: 100 μm). The results are presented as the mean ± SEM. Significant difference was estimated using an unpaired *t* test (* *p* < 0.05, ** *p* < 0.01, *** *p* < 0.001 vs. the control or the age-matched sham; # *p* < 0.05, ## *p* < 0.01 vs. 1w sham of each group).

**Table 1 cells-12-02412-t001:** GO enrichment analysis of genes up-regulated or down-regulated in R28.

GO Term	Definition	*p*-Value	No. up- or down Regulated	Total in Category
CoCl_2_ vs. Control				
Up				
GO:0007080	mitotic metaphase plate congression	9.36 × 10^−5^	5	111
GO:0006915	apoptotic process	1.68 × 10^−4^	13	111
GO:0005829	cytosol	9.57 × 10^−12^	66	115
GO:0005654	nucleoplasm	1.64 × 10^−8^	49	115
GO:0005634	nucleus	3.94 × 10^−8^	62	115
GO:0005874	microtubule	2.48 × 10^−7^	13	115
GO:0005737	cytoplasm	9.53 × 10^−6^	54	115
GO:0072562	blood microparticle	1.57 × 10^−4^	7	115
GO:0070062	extracellular exosome	4.06 × 10^−4^	26	115
GO:0072686	mitotic spindle	0.001097823	6	115
GO:0000776	kinetochore	0.00172263	6	115
GO:0005871	kinesin complex	0.002838309	4	115
GO:0005515	protein binding	6.20 × 10^−9^	102	113
GO:0008017	microtubule binding	2.60 × 10^−5^	10	113
GO:0005524	ATP binding	2.47 × 10^−4^	22	113
GO:0042802	identical protein binding	9.06 × 10^−4^	22	113
GO:0005198	structural molecule activity	0.001048323	7	113
GO:0031267	small GTPase binding	0.001514117	8	113
GO:0031625	ubiquitin protein ligase binding	0.00230867	8	113
Down				
GO:0005654	nucleoplasm	1.34 × 10^−4^	34	93
GO:0005829	cytosol	1.82 × 10^−4^	42	93
GO:0005515	protein binding	3.32 × 10^−5^	79	92
CoCl_2_ vs. CoCl_2_+NPC				
up				
GO:0031508	pericentric heterochromatin assembly	1.43 × 10^−4^	3	75
GO:0005829	cytosol	4.03 × 10^−4^	34	73
GO:0005515	protein binding	1.12 × 10^−4^	61	70
GO:0031492	nucleosomal DNA binding	4.49 × 10^−4^	4	70
Down				
GO:0072562	blood microparticle	5.96 × 10^−5^	6	60
GO:0070062	extracellular exosome	1.11 × 10^−4^	18	60
GO:0005737	cytoplasm	2.97 × 10^−4^	30	60
GO:0005829	cytosol	5.41 × 10^−4^	29	60
GO:0005634	nucleus	9.02 × 10^−4^	30	60
GO:0071682	endocytic vesicle lumen	0.001199461	3	60
GO:0005615	extracellular space	0.002971796	14	60
GO:0045095	keratin filament	0.003882598	4	60

## Data Availability

All data and materials are available upon request.

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
