# Peer review of "Unveiling Neuroprotection and Regeneration Mechanisms in Optic Nerve Injury: Insight from Neural Progenitor Cell Therapy with Focus on Vps35 and Syntaxin12"

_cells, 2023, doi:10.3390/cells12192412_

Round 1

Reviewer 1 Report

Axonal degeneration resulting from optic nerve damage can lead to the progressive death 16 of retinal ganglion cells (RGCs), culminating in irreversible vision loss. This paper demonstrates that ischemic injury and crush injury cause optic nerve damage via different mechanisms, which can be effectively simulated using ONCo and ONCr, respectively. Moreover, cell-based therapies such as NPCs may offer promising avenues for treating various optic neuropathies, including ischemic and crush injuries. The paper overall is interesting. Several comments are listed as follows,

(1)    The writing quality of the paper should be improved e.g.” with CoCl2, and with CoCl2 a”, “, previous our studies”->”, our previous studies” Several blank page in 11 and 16 should be removed.

(2)    I am not quite sure that how is the precision of the sample taken from the optic nerve. Will this influence the result?

(3)    Please consider to cite the paper” Cell-Based Biohybrid Sensor Device for Chemical Source Direction Estimation”

Axonal degeneration resulting from optic nerve damage can lead to the progressive death 16 of retinal ganglion cells (RGCs), culminating in irreversible vision loss. This paper demonstrates that ischemic injury and crush injury cause optic nerve damage via different mechanisms, which can be effectively simulated using ONCo and ONCr, respectively. Moreover, cell-based therapies such as NPCs may offer promising avenues for treating various optic neuropathies, including ischemic and crush injuries. The paper overall is interesting. Several comments are listed as follows,

(1)    The writing quality of the paper should be improved e.g.” with CoCl2, and with CoCl2 a”, “, previous our studies”->”, our previous studies” Several blank page in 11 and 16 should be removed.

(2)    I am not quite sure that how is the precision of the sample taken from the optic nerve. Will this influence the result?

(3)    Please consider to cite the paper” Cell-Based Biohybrid Sensor Device for Chemical Source Direction Estimation”

Author Response

Answers to the reviewers’ comments

I read all the comments and rewrote the phrases according to your suggestions.And I also tried to explain the ideas and answer to every question from you. I really appreciate your professional comments and excellent points to improve my manuscript for the readers.

Reviewer #1

Axonal degeneration resulting from optic nerve damage can lead to the progressive death 16 of retinal ganglion cells (RGCs), culminating in irreversible vision loss. This paper demonstrates that ischemic injury and crush injury cause optic nerve damage via different mechanisms, which can be effectively simulated using ONCo and ONCr, respectively. Moreover, cell-based therapies such as NPCs may offer promising avenues for treating various optic neuropathies, including ischemic and crush injuries. The paper overall is interesting. Several comments are listed as follows,

(1)    The writing quality of the paper should be improved e.g.” with CoCl2, and with CoCl2 a”, “, previous our studies”->”, our previous studies” Several blank page in 11 and 16 should be removed.

=> We rewrote the phrases to make it clear.

(2)    I am not quite sure that how is the precision of the sample taken from the optic nerve. Will this influence the result?

=> I concur with your viewpoint on the critical importance of preserving the optic nerve integrity for the purposes of this study. Allow me to elucidate the technique employed to achieve the intact removal of optic nerves from the animal subjects. Initially, we conducted the removal process by following these steps:

  1. Carefully removed the skin and fur covering the skull, exposing the underlying tissue.
  2. Inserted the tips of small scissors into the exposed area at the rear of the brainstem, which is characterized by its stark white appearance.
  3. While maintaining the scissors at an upward angle, we made a precise incision along the skull, starting from the bregma region and extending towards the frontal part of the brain.
  4. Utilized forceps to delicately peel and detach the skull, thereby revealing the brain.
  5. Subsequently, by gently elevating the brain with forceps, we confirmed the location of the optic nerve beneath the skull.
  6. Employing forceps with precision, we methodically removed the remaining bone surrounding the eye socket, separating it from the eye and optic nerve.
  7. Finally, we executed the optic nerve dissection as near as possible to the posterior section of the eye, ensuring a precise cut close to the optic chiasm (indicated by the yellow dotted line).

This meticulous procedure allowed for the retrieval of intact optic nerves, a fundamental requirement for our investigative purposes.

(3)    Please consider to cite the paper” Cell-Based Biohybrid Sensor Device for Chemical Source Direction Estimation”

=> I added the following sentences to the ‘Discussion’ section.

: To gain deeper insights into the role of syntaxin or delphilin in neural processes and the associated signaling mechanisms, the utilization of a cell-based biohybrid sensor device for chemical source direction estimation becomes instrumental. This innovative device not only aids in the exploration of cell functionality but also provides directional guidance towards chemical sources within retina progenitor cells like R28 cells, facilitating precise material analysis involved in signal transmission [29].

Reviewer 2 Report

It’s an interesting study and the findings are exciting. Authors need to address below mentioned points: 

The authors may need to clarify the rationale of comparing ONCo and ONCr models? What are the disease relevance and translational significance? Please also clarify the pathophysiological cascades in both models.

It’s interesting to see the comparisons at the mitochondrial, ganglionic, and axon level. Recent studies have shown that neuroimmunometabolism and immunometabolism are crucial in nerve regeneration following injury. It will be interesting to see any such evidence in this comparison.

Figure 2: in all immunostained images, scale bars are either missing or not visible. Please provide that.

Figure 4 and 5: Please provide uncropped full blot images as supplementary figures. Show the molecular weight markers for each.

Detail statistics (method of analysis) should be described in the method or each figure legend.

Good

Author Response

Answers to the reviewers’ comments

I read all the comments and rewrote the phrases according to your suggestions.And I also tried to explain the ideas and answer to every question from you. I really appreciate your professional comments and excellent points to improve my manuscript for the readers.

Reviewer #2

It’s an interesting study and the findings are exciting. Authors need to address below mentioned points:

The authors may need to clarify the rationale of comparing ONCo and ONCr models? What are the disease relevance and translational significance? Please also clarify the pathophysiological cascades in both models.

=> To encapsulate the findings of the national survey on optic neuropathy in Korea, this study delves into various optic neuropathies, with a particular focus on traumatic optic neuropathy (TON), nonarteritic anterior ischemic optic neuropathy (NAION), and Leber’s hereditary optic neuropathy (LHON). In the context of NAION, the use of steroids has been explored, primarily aimed at reducing axonal swelling to enhance optic nerve circulation. However, the efficacy of steroids in NAION remains a topic of debate. Notably, a US study by Atkins et al. revealed that among 350 neuro-ophthalmologists surveyed, 27% adopt an observational approach, 96% prescribe aspirin for preventive measures, while none opt for intravenous (IV) steroids (Graefe's Archive for Clinical and Experimental Ophthalmology, 2020, 258:1975–1981).

Despite the prevalence of ischemic neuropathies like NAION, the utilization of steroids remains limited, making this study a valuable reference for potential future clinical applications and a source of pertinent information for the clinical domain.

It’s interesting to see the comparisons at the mitochondrial, ganglionic, and axon level. Recent studies have shown that neuroimmunometabolism and immunometabolism are crucial in nerve regeneration following injury. It will be interesting to see any such evidence in this comparison.

=> I truly agree with you that understanding the role of neuronal immuno-metabolism in optic nerve recovery opens up new avenues for therapeutic interventions. Targeting specific metabolic pathways in neurons or immune cells may help shift the balance towards a pro-regenerative immune response. Additionally, promoting metabolic resilience in neurons can enhance their ability to withstand injury and initiate regeneration.

Therefore, we described the importance of neuronal immuno-metabolism in optic nerve recovery. For instance, Sema3A-Nrp1 signaling influencing microglial activation in the injured optic nerve was additionally explained in the discussion.

Figure 2: in all immunostained images, scale bars are either missing or not visible. Please provide that.

=> The scale bars were marked on Figure 2AB. The H&E image in Figure 2c is also shown with a scale bar on the sham of the 4w ONCr.

Figure 4 and 5: Please provide uncropped full blot images as supplementary figures. Show the molecular weight markers for each.

=> The molecular weight of target proteins were indicated on the un-cropped full blot images and we uploaded this as supplementary figures.

Detail statistics (method of analysis) should be described in the method or each figure legend.

=> I revised sentences of ‘Figure legends’ as follow: The results are presented as a mean ± SEM. Significantly difference was estimated using an unpaired t-test.
